# Cross-lingual Transfer Learning for Intent Detection of Covid-19 Utterances

**Abhinav Arora**[*]
Facebook

**Akshat Shrivastava**[*]
Facebook

**Mrinal Mohit**
Facebook

**Lorena Sainz-Maza Lecanda**
Facebook

**Ahmed Aly**
Facebook

## Abstract

In times of a global pandemic, interactive chat bots are an indispensable tool to provide information to people. With this motivation, we study the problem of intent detection of user utterances, which is usually the first language understanding step in such systems. Specifically, we focus on cross-lingual transfer learning for intent detection of user utterances and zero-shot learning for code-switched (CS) utterances. We release a multilingual dataset, M-CID, containing 6871 utterances across English, Spanish, French, German and Spanglish (Spanish + English). We use this dataset to explore some cross-lingual transfer learning techniques to study: (1) monolingual and multilingual model baselines, (2) cross-lingual transfer from English to Spanish, French and German, and (3) zero-shot code-switching for Spanglish. In our experiments, we observe that XLM-R models are able to significantly outperform cross lingual word embedding techniques for all of the above settings. We also show that it is possible to obtain a strong performance on code-switched data by only using monolingual data from substrate languages.

## 1 Introduction

In the wake of the Covid-19 crisis, it is of paramount importance to build interactive tools that can provide essential information such as Covid symptoms, treatment options, etc. These could either be information retrieval systems that fetch relevant articles (Zhang et al., 2020; Esteva et al., 2020; MacAvaney et al., 2020) or they could be interactive chat bots (WHO, 2020; Martin et al., 2020) that users can interact with. In this work, we explore the problem of intent classification; which is the first step of a natural language understanding system. For example, for an utterance such as *What*

are the indicators of covid infection?, the first step in responding to this request, is to identify that the user's intent is to ask for Covid-19 symptoms.

While neural models dominate intent prediction (Liu and Lane, 2016; Zhang and Wang, 2016; Zhang et al., 2018) they require a lot of training data. Consequently, developing these systems for many new languages can be a highly resource-intensive task, especially during global pandemic situations, internationalization is needed in a very short amount of time. Furthermore, multilingual systems often also need to support code-switching (CS), which is the alternation of languages within an utterance (Poplack, 2004). Collecting CS data is even harder as it requires bilingual annotators and the number of CS pairs grows quadratically with languages. Thus, there is a need to explore techniques that enable transfer learning from one or more languages to other languages and CS dialects.

In order to further study multilingual intent detection for Covid-19, **we release *M-CID (Multi-ingual Covid Intent Detection)*,** an open source intent detection dataset for Covid-19 chat bots. M-CID contains 6871 utterances across 16 intents for 4 languages: English, Spanish, French, and German along with a Spanglish test set for CS. **We provide several strong baselines** to show the impact of cross lingual embedding such as MUSE (Conneau et al., 2017), SentencePiece embeddings from XLM-R (Kudo and Richardson, 2018; Conneau et al., 2020), aligning ELMo representations (Peters et al., 2018; Schuster et al., 2019) and pre-trained multilingual transformers, XLM-R (Conneau et al., 2020), comparing monolingual training against cross lingual training. On our dataset, we show that XLM-R models significantly out perform cross lingual embeddings and **cross lingual training improves performance across most models compared to monolingual training.** In addition **we also show the impact of cross lingual trans-**

---
[*] Correspondence to {`abhinavarora,akshats`} `@fb.com`

|        | EN   | ES   | FR   | DE   | Spanglish |
|--------|------|------|------|------|-----------|
| Train  | 1258 | 1106 | 1105 | 1086 | 0         |
| Eval   | 148  | 161  | 173  | 188  | 0         |
| Test   | 339  | 333  | 315  | 326  | 333       |
| Total  | 1745 | 1600 | 1593 | 1600 | 333       |

Table 1: Summary statistics of the dataset. Note that *Spanglish* only has a test set for zero-shot evaluations.

**fer learning**, where we train with the full English train set and small portions of other languages, and also show strong zero-shot transfer with XLM-R based models. Lastly, we study the impact on our code switching test set and show that **monolingual training on English and Spanish for XLM-R based models is sufficient for code switching**.

## 2 Data

We release M-CID, a dataset of **6871** natural language utterances across **16** Covid-19 specific intents and **4** languages: English, Spanish, French and German. Additionally, the dataset also contains a Spanglish test set for CS evaluation. All of these utterances were synthetically created by annotators based on an ontology describing all intents with few representative examples. No user data was used in this process. Monolingual utterances were authored by native speakers using the described ontology and Spanglish utterances were created by one of the authors, who is bilingual in Spanish and English.

We believe that this data provides a great opportunity to explore cross-lingual classification for Covid-19 chat bots and to the best of our knowledge, this is the first multilingual dataset for an intent detection task for Covid-19 utterances. Table 1 contains the utterance counts for each language across the training, evaluation and test splits. More details about the intent labels, distribution of utterances across them, and some representative examples are presented in Appendix A.

We release the data at `https://fb.me/covid_mcid_dataset`.

## 3 Modeling Approaches

In the following section, we provide a brief description of all the models and the implementations used. We use accuracy as our evaluation metric, which works well for our setup because the intent labels have a balanced distribution in the dataset. Appendix C contains details regarding reproducibility and model hyperparameters for further reference.

| Model | Setting | Accuracy | | | |
|-------|---------|-------|-------|-------|-------|
|       |         | EN    | ES    | FR    | DE    |
| MUSE  | Mono    | 81.12 | 76.28 | 69.52 | 80.06 |
|       | XL      | 81.12 | 78.98 | 69.21 | 82.82 |
| SP    | Mono    | 82.89 | 79.58 | 73.97 | 81.9  |
|       | XL      | 83.48 | 84.08 | 77.14 | 86.50 |
| ELMo  | Mono    | 86.14 | 84.98 | 76.83 | 84.05 |
|       | XL      | 87.61 | 88.29 | 80.95 | 85.28 |
| XLM-R Base | Mono | 90.27 | 88.59 | 87.30 | 89.88 |
|       | XL      | 89.97 | 92.19 | 87.94 | 92.64 |
| XLM-R Large | Mono | **91.45** | 91.29 | 88.25 | **92.94** |
|       | XL      | 91.15 | **93.69** | **89.52** | 92.94 |

Table 2: Full training results for all languages. *Mono* refers to a monolingual model for each language and *XL* refers to a shared multilingual model.

### 3.1 Cross-Lingual Word Embeddings

Our base model is a CNN based text classification model based on the architecture described by Kim (2014). For regularization, we add a dropout (Srivastava et al., 2014) after the convolution and pooling layers. In order to enable language transfer, we use pre-trained cross-lingual word embeddings as an input to the model. We experiment with the following embedding strategies:

- **MUSE**: We use MUSE word embeddings (Conneau et al., 2017), with a vocabulary size of 25K of for all the three languages. These are fastText (Bojanowski et al., 2017) Wikipedia supervised word embeddings, aligned in a single vector space. We refer to this model as simply MUSE.

- **SentencePiece Embeddings**: We experiment with pre-trained SentencePiece embeddings obtained from a large multilingual corpus. Specifically, we use the SentencePiece (Kudo and Richardson, 2018) tokenization and take the embedding values from the already-trained XLM-R (large) (Conneau et al., 2020) weights. Since these are sub-word embeddings, they tend to be robust to misspellings and rare tokens by breaking them down into better-known sub-tokens. We refer to this model as simply SP.

- **Cross-lingual ELMo**: We also experiment with aligned multi-lingual deep contextual embeddings obtained by aligning monolingual ELMo embeddings (Peters et al., 2018). We use the ELMo models and alignments released

| Model | Spanish % Training | | | | | French % Training | | | | |
|---|---|---|---|---|---|---|---|---|---|---|
| | Zero-shot | 10 | 20 | 50 | 80 | Zero-shot | 10 | 20 | 50 | 80 |
| MUSE (F) | 59.76 | 63.66 | 67.27 | 66.97 | 72.37 | 47.30 | 54.6 | 60.95 | 60.63 | 66.03 |
| SP (F) | 33.03 | 66.67 | 72.97 | 79.58 | 81.98 | 29.84 | 59.47 | 66.67 | 69.84 | 77.46 |
| MUSE | 25.83 | 52.85 | 61.86 | 69.37 | 75.68 | 24.76 | 40.00 | 55.87 | 60.32 | 70.48 |
| SP | 38.74 | 55.56 | 60.36 | 69.07 | 75.38 | 29.52 | 47.62 | 54.60 | 59.68 | 70.48 |
| ELMo | 71.17 | 75.68 | 83.78 | 82.88 | 88.59 | 63.17 | 65.71 | 73.02 | 73.97 | 79.37 |
| XLM-R Base | 84.98 | 86.49 | 90.69 | 90.99 | 93.09 | 78.73 | 82.86 | 86.03 | 86.35 | **89.52** |
| XLM-R Large | **90.99** | **90.39** | **91.29** | **92.79** | **93.39** | **83.17** | **83.17** | **86.98** | **87.94** | 88.25 |

Table 3: Results for cross-lingual transfer for all models. *(F)* refers to freezing the embeddings during training. In the zero-shot setting, only English data is used for training and model selection. For others, the specified percentage of target training data is also used along with English.

by Schuster et al. (2019). Specifically, we use the alignments of the first LSTM layer, which the authors found best in their experiments.

### 3.2 Pre-trained Cross-Lingual Language Models

Using the same accuracy metric as above, we also examine the performance of pre-trained XLM-R (Conneau et al., 2020) models. These models are pre-trained via an unsupervised Masked Language Modeling (MLM) objective (Devlin et al., 2019) on massive multilingual data. They share a Sentence-Piece representation and a common transformer encoder (Vaswani et al., 2017) for different languages. In order to use this for intent classification, we add a linear classifier on top of the first hidden state of the Transformer and fine-tune the network on our dataset. For our experiments, we report results with both XLM-R Base and XLM-R Large which are pre-trained on 100 languages and are provided by the PyText framework (Aly et al., 2018).

**Results and Discussion** Table 2 shows the test set accuracy for all of the above models using the full training data. In the *mono* setting a model is trained per language using the data of only that language. In the *XL* setting a single cross-lingual model is trained using the data for all the languages together. For these experiments, MUSE and SP embeddings were not frozen during training. While we get different results for each language, there are several consistent patterns. XLM-R models significantly outperform other models. We also see that cross-lingual models trained with all the 4 languages mostly do better than their monolingual counterparts, barring few exceptions. Amongst the cross-lingual embeddings, SP embeddings are better than MUSE, which is expected as they operate

on subword units that are shared across languages. Aligned ELMo embeddings mostly perform better than both of these due their contextual nature.

## 4 Cross-lingual Learning

### 4.1 Language Transfer

In this set of experiments, we examine the language transfer abilities of our models. Specifically, we treat English as our source language, and Spanish, French and German as the target languages. For each of the models discussed above, we first run zero-shot experiments where only English data is used for training and model selection. We then run learning curve experiments, where we progressively sample 10, 20, 50 and 80 percent of the target language training data and upsample it so that it roughly matches the size of the English data. Here, model selection is done using the evaluation splits of all languages.

**Results and Discussions** Table 3 shows the cross-lingual transfer results for Spanish and French. From these results, it is evident that XLM-R large can achieve very strong performance for zero-shot transfer from English. For Spanish, the zero-shot performance is 2.4 absolute points lesser than using 80% Spanish training data. For French, this gap is higher and there is a progressive improvement from zero-shot to 80% training. For both the languages, we see that having target language training data yields better performance than zero-shot. XLM-R base follows a similar trend as large. Interestingly, for French, XLM-R base has slightly better results compared to XLM-R large with 80% training data, which can be attributed the high sensitivity of XLM-R fine-tuning to learning rate.

| Model | Setting | | |
|---|---|---|---|
| | EN | ES | EN + ES |
| MUSE (F) | 63.06 | 48.65 | 70.57 |
| SP (F) | 62.76 | 43.24 | 78.38 |
| MUSE | 69.67 | 42.94 | 76.88 |
| SP | 68.77 | 55.86 | 79.88 |
| XLM-R Base | 83.78 | 77.78 | 88.29 |
| XLM-R Large | **87.39** | **91.29** | **88.89** |

Table 4: Zero-shot code-switching results for each of the training settings. *(F)* refers to freezing the embeddings during training.

For MUSE and SP, we show results with both freezing and fine-tuning the embeddings during training. For MUSE, we find that freezing the word embeddings yields a significantly better performance compared to fine-tuning in the lower resource settings (<50%), as the model does not overfit to the source language. For SP, freezing the embeddings is better than fine-tuning in most settings. This can be attributed to the overlap of subwords across languages. Similar to table 2, we generally observe better language transfer with SP as compared to MUSE. Similarly, contextual ELMo embeddings perform better than both of these. Compared to XLM-R, all of these approaches have a much bigger performance gap between zero-shot and 80% target language training. This suggests that XLM-R is very effective at zero-shot cross-lingual transfer, which aligns with the findings of Wu and Dredze (2019).

Appendix B discusses cross-lingual transfer results for German, which exhibits similar patterns as Spanish and French, as discussed above.

## 4.2 Zero-shot Code-Switching

Since code-switching is a big part of spoken language in many cultures, we also investigate the performance of our models on Spanglish, which is a mix of English and Spanish. These are zero-shot experiments where we neither use CS data for model training nor for model selection. The only data available is monolingual English and Spanish data. For each of our models discussed above, we experiment with three training data settings. We first train two models using the training data of each of the two languages one by one, and then a model using both Spanish and English data.

**Results and Discussions** Table 4 shows the zero-hot CS performance of different models. We do

not perform ELMo experiments for CS as it is not intuitive to represent Spanglish context with monolingual ELMo. From the results, we can see that XLM-R models perform very well even when fine-tuned on English only or Spanish only. XLM-R large fine-tuned on Spanish only, outperforms all other model settings. We also see that for MUSE and SP, training on English only gives better performance than Spanish only setting. We believe this is because for Spanglish utterances, the trigger words such as *treatment*, *vaccine*, *donation*, etc are usually in English and thus the English only model is able to do well. Further, freezing the embeddings is usually worse for all settings.

## 5 Related Work

**Cross-lingual Transfer Learning** Majority of the initial work on cross-lingual transfer was centered around aligning pre-trained word embeddings to a common vector space (Xing et al., 2015; Zhang et al., 2017; Conneau et al., 2017). Schuster et al. (2019) and Aldarmaki and Diab (2019) further build on this by exploring context-aware cross-lingual alignment of contextualized representations from ELMo (Peters et al., 2018). More recently, pre-trained multilingual masked language models such as mBERT (Devlin et al., 2019), XLM (Lample and Conneau, 2019) and XLM-R (Conneau et al., 2020) have been introduced. XLM-R obtains state-of-the-art performance on the XNLI (Conneau et al., 2018) benchmark.

## 6 Conclusion

In this paper, we release M-CID, a dataset for mulitilngual Covid-19 intent detection across English, Spanish, French, German and Spanglish. We provide several baselines to show the impact of various cross lingual representations and pre-trained transformers on this dataset, along with a zero-shot, few-shot and code-switching studies of cross lingual transfer for intent detection. We show XLM-R based models provide very strong baselines compared to cross lingual embedding models. We hope that the release of M-CID will allow for further research for cross lingual intent detection in Covid chat bots.

## Acknowledgments

We thank Caitlin Lohman, Claire Lesage and Zainab Hossainzadeh for driving data collection.

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

# Appendix

## A Dataset Details

As an extension of table 1, we show the intent distribution across languages and across train, eval, and test split in table 5.

## B German Cross-lingual Transfer

| Model | German % Training | | | | |
| | Zero-shot | 10 | 20 | 50 | 80 |
| --- | --- | --- | --- | --- | --- |
| MUSE (F) | 49.08 | 66.26 | 68.40 | 73.93 | 78.53 |
| SP (F) | 33.13 | 66.87 | 75.15 | 78.22 | 80.67 |
| MUSE | 23.62 | 50.92 | 63.50 | 71.17 | 79.45 |
| SP | 33.84 | 61.04 | 67.79 | 75.77 | 80.67 |
| ELMo | 61.04 | 69.33 | 77.91 | 80.98 | 83.44 |
| XLM-R Base | 83.74 | 85.28 | 89.88 | 91.10 | 91.41 |
| XLM-R Large | **88.34** | **89.26** | **90.80** | **92.02** | **91.72** |

Table 6: Results for cross-lingual transfer experiments for German, similar to the Spanish and French experiments shown in Table 3.

Table 6 shows the cross-lingual transfer results for German similar to the results for Spanish and French in Table 3. We see similar patterns for German as for Spanish in Section 4.1. As expected, XLM-R large achieves the best zero-shot performance and is very close to the performance with 80% target language training data. For all models, we see that having target language training data yields better performance than zero-shot. Similar to Spanish and French, we find that freezing the word embeddings yields a significantly better performance compared to fine-tuning in the lower resource settings for MUSE and in most settings for SP. Further, aligned ELMo provides better cross-lingual transfer than both SP and MUSE due to the contextual nature of the embeddings.

## C Hyperparameters for Models

We detail the experimental set up for each of our models below. For hyperparameter tuning, we sweep over the learning rate and batch size across model architectures.

**Baseline DocNN Model**  For all of our DocNN experiments we keep the DocNN model architecture consistent and sweep the learning rate and batch size. Here we detail the architecture. We use a CNN model with kernel sizes [3,4,5] and 100 feature maps per kernel. We employ dropout (Srivastava et al., 2014) of 0.25. We then add an MLP with hidden dimension 128 to project to the output classes. We optimize for the cross entropy loss, and leverage the AdamW optimizer (Loshchilov and Hutter, 2017). All our models are trained across 8 GPUs using distributed data parallel training with PyTorch (Paszke et al., 2019). Our effective batch size is computed by multiplying the batch size per worker by the number of workers.

**MUSE DocNN**  We initialize our embedding layer with 300 dimension MUSE embeddings. We train for 100 epochs with an effective batch size of 512 and learning rate 0.000691 for cross lingual, 256 and 0.00135 for English, 256 and 0.000876 for Spanish, 256 and 0.00135 for French, 512 and 0.00233 for German.

**Frozen MUSE DocNN**  We use the same setup as the MUSE DocNN model however, noteably we freeze the MUSE embeddings. We train 100 epochs and use 256 batch size with a learning rate of 0.001345 for cross lingual, English, Spanish, French, and German.

**SentencePiece (SP) DocNN**  We use sentence piece embeddings loaded from the XLM-R Large model with embedding dimension 1024. We use an effective batch size of 256 and learning rate 0.00178 for cross lingual, English, Spanish, French, and German.

**Frozen SP DocNN**  We use the same configeration as SP DocNN, however we freeze the sentence piece embeddings. We use an effective batch size of 512 and learning rate 0.000217 for cross lingual, English, Spanish, French, and German.

**Cross-lingual ELMo DocNN**  We use ELMo embeddings from AllenNLP (Gardner et al., 2017) and get 1024 dimension aligned embedding representations using the alignments released by Schuster et al. (2019). We train 100 epochs with an effective batch size of 256 and learning rate of 0.000592 for cross lingual training, 256 and 0.00115 for English, 256 and 0.00115 for Spanish, 512 and 0.00222 for French, 256 and 0.000216 for German.

**XLM-R Base**  We train our XLM-R base models for 40 epochs with an effective batch size of 512. We leverage the Adam (Kingma and Ba, 2014) optimizer, and use a learning rate of 0.00005 for cross lingual training, 0.000075 for English monolingual training, 0.00005 for Spanish monolingual training, 0.000075 for French monolingual training, 512 and 0.00005 for German monolingual training.

**XLM-R Large**  Similar to XLM-R Base we train our models for 40 epochs, we leverage an effective batch size of 128. We use the Adam optimizer, and use a learning rate of 0.00005 for cross lingual training, 0.00002 for English monolingual training, 0.00001 for Spanish monolingual training, 0.00001 for French monolingual training, and 0.00002 for German monolingual training.

| Intent | Split | Number of Occurrences | | | | |
|---|---|---|---|---|---|---|
| | | English | Spanish | French | German | Spanglish |
| what_is_corona | Train | 82 | 73 | 71 | 70 | - |
| *"what is coronavirus"* | Eval | 6 | 15 | 12 | 9 | - |
| *"can you tell me about the virus"* | Test | 22 | 12 | 17 | 21 | 15 |
| what_if_i_visited_high_risk_area | Train | 72 | 68 | 71 | 66 | - |
| *"i traveled to new york recently am i infected"* | Eval | 9 | 8 | 8 | 10 | - |
| *"how do i protect myself in high risk areas"* | Test | 24 | 24 | 21 | 24 | 25 |
| what_are_treatment_options | Train | 92 | 70 | 60 | 65 | - |
| *"do we have a cure yet"* | Eval | 9 | 8 | 8 | 16 | - |
| *"do hospitals know how to fix this"* | Test | 24 | 24 | 21 | 19 | 25 |
| what_are_symptoms | Train | 72 | 66 | 75 | 72 | - |
| *"i have a cold should i be worried"* | Eval | 15 | 16 | 8 | 7 | - |
| *"is coughing a sign of the virus"* | Test | 23 | 18 | 17 | 21 | 21 |
| travel | Train | 87 | 63 | 71 | 64 | - |
| *"is it safe to travel now"* | Eval | 5 | 11 | 10 | 13 | - |
| *"can i take the bus to work"* | Test | 18 | 26 | 19 | 23 | 26 |
| share | Train | 82 | 67 | 62 | 66 | - |
| *"share this with jack"* | Eval | 9 | 12 | 11 | 10 | - |
| *"send this info to my friends"* | Test | 19 | 21 | 27 | 24 | 24 |
| protect_yourself | Train | 76 | 68 | 72 | 75 | - |
| *"how can i stay safe"* | Eval | 14 | 15 | 8 | 9 | - |
| *"what should i do to prevent"* | Test | 20 | 17 | 20 | 16 | 25 |
| okay_thanks | Train | 71 | 70 | 61 | 69 | - |
| *"thanks for doing this"* | Eval | 13 | 9 | 16 | 12 | - |
| *"this is amazing"* | Test | 26 | 21 | 16 | 19 | 7 |
| news_and_press | Train | 80 | 73 | 73 | 71 | - |
| *"what's the latest"* | Eval | 8 | 7 | 11 | 13 | - |
| *"did anything big happen today"* | Test | 22 | 20 | 16 | 16 | 26 |
| myths | Train | 70 | 68 | 75 | 69 | - |
| *"what are myths about covid"* | Eval | 8 | 7 | 10 | 12 | - |
| *"what are the misconceptions"* | Test | 32 | 24 | 15 | 19 | 21 |
| latest_numbers | Train | 78 | 74 | 68 | 64 | - |
| *"what's the latest statistics"* | Eval | 7 | 9 | 10 | 15 | - |
| *"what do the numbers look like now"* | Test | 25 | 17 | 22 | 21 | 24 |
| how_does_corona_spread | Train | 81 | 71 | 64 | 68 | - |
| *"how does the virus spread"* | Eval | 9 | 7 | 10 | 9 | - |
| *"can people with masks transmit to other people"* | Test | 20 | 22 | 26 | 23 | 24 |
| hi | Train | 82 | 74 | 67 | 71 | - |
| *"hello"* | Eval | 9 | 8 | 15 | 14 | - |
| *"hey covid bot"* | Test | 19 | 18 | 18 | 15 | 7 |
| donate | Train | 81 | 67 | 75 | 64 | - |
| *"this is great how do i help you"* | Eval | 9 | 8 | 8 | 13 | - |
| *"i wish i could do something about this"* | Test | 20 | 25 | 17 | 23 | 20 |
| can_i_get_from_packages_surfaces | Train | 73 | 71 | 69 | 69 | - |
| *"is it safe to get food delivered"* | Eval | 8 | 7 | 11 | 10 | - |
| *"how often should i clean my table"* | Test | 24 | 22 | 20 | 21 | 25 |
| can_i_get_from_feces_animal_pets | Train | 79 | 62 | 71 | 63 | - |
| *"can i get the virus from dogs"* | Eval | 10 | 13 | 7 | 16 | - |
| *"should i stop eating meat"* | Test | 16 | 25 | 22 | 21 | 21 |

Table 5: Dataset details by intent labels. For each intent listed are the occurrences of each label in the train, eval, and test set by language. Italicised underneath each label are two samples of utterances for that intent. *Note: Spanglish is only available as a test set hence there are no training or validation samples*