# OpenReview forum: "Cross-lingual Transfer Learning for Intent Detection of Covid-19 Utterances"
_EMNLP/2020/Workshop/NLP-COVID — Submitted to NLP-COVID19-EMNLP_

### Official Review · AnonReviewer3 · 2020-09-21
**Interesting contribution, needs minor clarifications**

**Rating:** 4
**Confidence:** 3

**Review:**

This paper introduces a multilingual dataset of 6871 utterances across 4 language + a mixture of English and Spanish. The dataset itself is an interesting contribution for the research community, which the authors complement with the study of different cross-lingual transfer learning techniques. I consider the paper to be appealing for the audience of the workshop, however it is not clear whether it is a short or a long one. I am considering it a short one, although the authors use the "annex escape" to complement important aspects of the paper.

In terms of methodology, could the authors elaborate more on the choice of the 16 intents? What was the basis for chosing them over others?Also, I would have liked to have more details on the ontology they mention in the beginning of Section 2. The C Section of the Appendix is a bonus point in terms of reproducibility of the results. However, I do not understand Table 6 in Section B of the same Appendix. Why was German considered less relevant than the results in Spanish and French (Table 3). Table 5 is quite explanatory and an important part of the paper. However, I would ask for the authors to elaborate more on the distinction between some of the intent categories, that seem quite similar. For example, "I traveled to new york recently" could also be understood as "travel".

---

> ### Author Response · Authors · 2020-09-28
> **Response to AnonReviewer3**
>
> We thank the reviewer for their very helpful feedback and comments.  We respond to the following concerns from the reviewer in the same order as listed by them in the review:
>
> 1. The choice of these 16 intents comes from a careful examination of the Whatsapp bot built by World Health Organization for answering common questions related to Covid -19 (https://www.who.int/news-room/feature-stories/detail/who-health-alert-brings-covid-19-facts-to-billions-via-whatsapp). We examined the menu driven options of this bot to build intents such as 'news_and_press', 'latest_numbers', 'myths', 'travel',  'donate', etc. Furthermore, we augmented this initial list of intents by examining the type of questions provided by this bot in its FAQ option. Combining both of these sources, we compiled the list of these 16 intents. In Section 2, by ontology we refer to these 16 intent types and the types of utterances that they represent.
>
> 2. In Table 3, we omit results from German in the interest of space. German is equally as relevant as French and Spanish. However, German exhibited exactly the same trends as Spanish and hence we skipped it to keep the main part of the paper within 4 pages.
>
> 3. We thank the reviewer for this thoughtful feedback. Intent 'travel' is primarily for travel related advice, whereas the intent 'what_if_i_visited_high_risk_area' is primarily catered towards past travel to an area or a location that the user feels is a Covid-19 hot spot.
>
> For the final version, we would be happy to add an additional page to the paper that covers more details on the dataset, intent choices as well as include the German results in the main paper body.

---

### Official Review · AnonReviewer2 · 2020-09-21
**Interesting dataset and reasonable baselines**

**Rating:** 5
**Confidence:** 4

**Review:**

This work presents a multilingual dataset for intent detection of COVID-related utterances. Besides, the authors built various baselines and experimented with code-switched data (Spanglish).

**Pros**
- The dataset is open-sourced and could be helpful for further COVID-related development.
- Reasonable analysis and discussions on baseline results.
- Hyperparameters of models were provided.

**Cons**
- The dataset `is synthetically created by annotators based on an ontology describing all intents with few representative examples`. However, no details on how synthetically created, the information about annotators, the ontology and how it was created, and the seed examples were provided. There is also no discussion on how close the dataset is to real utterances. These issues make the dataset, i.e., the main contribution, less useful.


**Misc**
- For Table 3, it might be better to have a plot instead of a table to show the trend when increasing % training.

- The authors should discuss why zero-shot performance with XLM-R Large is better than using 10% Spanish data, and why it performs worse than the base model under the setting of 80% French data.

---

> ### Author Response · Authors · 2020-09-28
> **Response to AnonReviewer2**
>
> We thank the reviewer for their valuable feedback. We respond to the following concerns in the same order as listed by the reviewer:
> 1. In order to create the dataset, we first identified 16 intents that we wanted to cover. These 16 intents comes from a careful examination of the Whatsapp bot built by World Health Organization for answering common questions related to Covid -19 (https://www.who.int/news-room/feature-stories/detail/who-health-alert-brings-covid-19-facts-to-billions-via-whatsapp). We examined the menu driven options of this bot to build intents such as 'news_and_press', 'latest_numbers', 'myths', 'travel', 'donate', etc. Furthermore, we augmented this initial list of intents by examining the type of questions provided by this bot in its FAQ option. Combining both of these sources, we compiled the list of these 16 intents. After identifying these intent types, we manually created seed examples illustrating these intents. These seed examples and the intent descriptions were provided to native speakers (who we referred to as annotators), who were asked to generate diverse set of utterances corresponding to these intents.
>
> 2. Thank you for the useful feedback on Table 3. We agree that a plot would be more intuitive compared to the table and we would be happy to replace the table in the final version of the paper.
>
> 3. We believe that the zero shot of XLM-R large exhibits slightly better performance than 10% for Spanish due to the high variance of XLM-R fine-tuning and it's sensitivity to learning rate. The same is true for the comparison between XLM-R base and large for 80% French. In the final version of the paper, we would certainly elaborate this more with results across multiple random seeds.

---

### Official Review · AnonReviewer1 · 2020-09-21
**useful task but needs more clarification on the dataset**

**Rating:** 4
**Confidence:** 5

**Review:**

This paper introduces a multilingual dataset for detecting COVID-19 specific intents and a Spanish test set for code-switching. The strength of this paper is that they introduced a dataset to the community, provided several baseline models in the experiments, and showed that using cross-lingual representation is useful in this task. However, there are some major problems about the dataset which the authors need to elaborate more on.

Pros:
* The paper presents to the community a public multilingual dataset for detecting covid-19 specific intents in user utterances.
* The paper shows that cross-lingual representation could improve the baseline models in this task.
* The paper is well written and provides relatively comprehensive comparison between baseline models. It also provides analysis about the performance of these models.

Cons:
* The dataset is synthetically created by annotators based on an ontology, but there is no description about the ontology. Furthermore, there is no discussion about how close this synthetic dataset is to real data. I wonder if the synthetic data could reflect the real world problem or it is just a much more simplified problem with utterances expressed in less diverse forms.
* The authors did not mention how and why the 16 intents are chosen, and also lack the definition of these intents. Is the formulation of these intents based on the real data or the synthetic data? Do most of the real data fall into these intent categories?
*  The authors need to elaborate more on the details of the annotation process, including the number of annotators to annotate each utterance and the inter-annotator agreement among annotators. How did they deal with a sample that received different annotation labels ( majority vote? adjudication among annotators? just throw them away? )? These details are important for other researchers to assess the quality of the dataset.

    In general the authors need to provide more details about the dataset. Otherwise it is hard to assess the quality of the dataset and makes it less useful to the community.

Some suggestions:
* I would suggest the authors give some significant test statistics when they claim the performance scores are significantly better.
* It would be interesting to see the breakdown of scores(Precision, Recall, F1) for each intent category and some analysis of the difficulty/easiness to identify each category. Showing cases with correct/incorrect predictions could also be helpful.

---

> ### Author Response · Authors · 2020-09-28
> **Response to AnonReviewer1**
>
> We thank the reviewer for the detailed review and their valuable feedback. We respond to the following concerns from the reviewer in the same order as listed in the review:
>
> 1. In the paper, by ontology we refer to the different types of intents and a description of what they mean. The 16 intents that we choose are obtained by a careful study of the Whatsapp bot built by World Health Organization for answering common questions related to Covid -19 (https://www.who.int/news-room/feature-stories/detail/who-health-alert-brings-covid-19-facts-to-billions-via-whatsapp). We examined the menu driven options of this bot to build intents such as 'news_and_press', 'latest_numbers', 'myths', 'travel', 'donate', etc. Furthermore, we augmented this initial list of intents by examining the type of questions provided by this bot in its FAQ option. Combining both of these sources, we compiled the list of these 16 intents. Since the basis for these 16 intents was the WHO bot, we believe that these 16 intents are reflective of real world use cases.
>
> 2. For data creation, we shared a description of these 16 intents and some seed examples with a pool of native speakers for each language. They were asked to generate diverse utterances corresponding to each of the 16 intents. The generated utterances were then finally vetted by a trained linguist who discarded or fixed any incorrect data.
>
> 3. Thank you for the valuable suggestions. We would certainly add some test statistics and a through breakdown of Precision and Recall scores for each intent. In the final version of the paper, we shall also add some common false positives for each category.